# Image Classifier for an Online Footwear Marketplace to Distinguish between Counterfeit and Real Sneakers for Resale

**DOI:** 10.3390/s24103030

**Published:** 2024-05-10

**Authors:** Joshua Onalaja, Essa Q. Shahra, Shadi Basurra, Waheb A. Jabbar

**Affiliations:** Faculty of Computing, Engineering and Built Environment, Birmingham City University, Birmingham B4 7RQ, UK; joshua.onalaja@mail.bcu.ac.uk (J.O.); shadi.basurra@bcu.ac.uk (S.B.)

**Keywords:** sneakers, SVM, Alex architecture, VGGI architecture, LeNet-5, bespoke architecture

## Abstract

The sneaker industry is continuing to expand at a fast rate and will be worth over USD 120 billion in the next few years. This is, in part due to social media and online retailers building hype around releases of limited-edition sneakers, which are usually collaborations between well-known global icons and footwear companies. These limited-edition sneakers are typically released in low quantities using an online raffle system, meaning only a few people can get their hands on them. As expected, this causes their value to skyrocket and has created an extremely lucrative resale market for sneakers. This has given rise to numerous counterfeit sneakers flooding the resale market, resulting in online platforms having to hand-verify a sneaker’s authenticity, which is an important but time-consuming procedure that slows the selling and buying process. To speed up the authentication process, Support Vector Machines and a convolutional neural network were used to classify images of fake and real sneakers and then their accuracies were compared to see which performed better. The results showed that the CNNs performed much better at this task than the SVMs with some accuracies over 95%. Therefore, a CNN is well equipped to be a sneaker authenticator and will be of great benefit to the reselling industry.

## 1. Introduction

Sneakerhead subculture started in the late 1980s in the United States and went global by the late 1990s. In the past, diehard sneaker collectors used to queue outside outlets and go to sneaker events in search of rare, “deadstock” (pairs of unworn sneakers), vintage, and limited-edition trainers [1]. Currently, the ways consumers shop for sneakers have drastically evolved due to increased technological advancements in internet accessibility and smartphone applications [2]. Now, they can be bought on a first-come-first-served basis from online retailers like Nike, Adidas, Footlocker, Footasylum, Offspring, and more. This allows companies to communicate directly with their customers, to build relationships, and to control brand messaging and identity [3]. The most sought-after sneakers are usually sold via online raffles through Nike’s “SNKRS” phone app and Adidas’s “Confirmed” app, where winners are chosen at random. These apps were made to give everyday buyers a better chance to purchase exclusive pairs, but they have delivered varied results [4]. One reason for this is these apps’ algorithms, which are thought to push the bulk of their limited-edition trainers to the most devoted and engaged customers. Another is the extreme disparity between the number of available pairs and the amount of people trying to buy them. This makes sneaker shopping a thrilling experience that drives consumer demand, which is what brands and retailers want [5].

Artificial intelligence is revolutionizing the realm of footwear recognition with remarkable precision and efficiency. Through advanced algorithms and deep learning models, AI systems can swiftly identify and classify various types of shoes based on their design, color, texture, and brand [6]. By harnessing convolutional neural networks (CNNs) and image recognition techniques, these systems can analyze intricate patterns and features, enabling accurate categorization of sneakers, sandals, boots, and more [7]. Moreover, AI-powered shoe recognition offers multifaceted applications, spanning from retail inventory management to enhancing personalized shopping experiences online [8]. With the ability to swiftly process vast amounts of visual data, AI empowers businesses to streamline operations, optimise supply chains, and deliver tailored recommendations to consumers [9]. As AI continues to evolve, its role in revolutionizing shoe recognition underscores its transformative impact across diverse industries [10]. Naturally, any form of human verification is inefficient and in huge companies that see millions of products a month it is difficult if not impossible not to make mistakes in the process of verifying the legitimacy of a pair of sneakers [11]. Another disadvantage of human sneaker authentication is the length of time it takes to train employees to perform the role and gain experience with it. Moreover, this middle-man approach causes delays in the buying and selling process and raises environmental concerns because products are shipped to and from multiple places [12]. An electronic sneaker authenticator using a computer algorithm can potentially help to avoid these issues. Utilising artificial intelligence for automatic sneaker authentication could be performed by using an image classifier as suggested by [13]. It should save ecommerce platforms time by speeding up sneaker authentication as well as enhancing their reputation in being known as resellers of genuine sneakers. It would also improve customer satisfaction as buyers can obtain their sneakers quicker and sellers can receive payment faster. Therefore, using artificial intelligence to be able to accurately and efficiently classify whether a sneaker being resold is genuine or fake would be invaluable to an online platform. Moreover, it would benefit the environment, which is now more important than ever in the sneaker industry according to [14]. StockX has recently made use of artificial intelligence to aid their authentication process, but they have not revealed which machine learning algorithms were employed.

This paper aims to design and develop an image classifier to help the Sole Suppliers sneaker authenticators to distinguish between real and counterfeit sneakers so less human inspection is required.

The organization of this paper is as follows: Section 2 reviews the most relevant research of using different image classifiers for many related applications. Section 3 describes the research methodology and how it is investigated in more detail. Section 4 and Section 5 present the numerical results and a discussion of these results. Finally, Section 6 concludes the work and shows the future work.

## 2. Literature Review

Nowadays, the use of artificial intelligence (AI) in the fashion world to classify images of clothing is becoming more prevalent as it has numerous benefits and applications according to Prajapati et al. [15]. These include arranging the collections of a fashion firm and recommending customers specific clothes. In the literature, many image classifiers have experimented with the widely available MNIST and Fashion-MNIST datasets.

For instance, Greeshma and Sreekumar [16] used an SVM classifier alongside the feature extractors histogram of oriented gradient and local binary patterns to classify the 10 groups of images in the Fashion-MNIST dataset. This gave an accuracy of 87.4%, which was 2.8% higher than Zhang et al. [17], who used linear support vector classification without any feature extraction, and 0.87% more than [16], who used an SVM classifier with only a histogram of the oriented gradient. This also highlights the small but significant difference that feature extraction can make to a classifier. Nonetheless, the predominant models used for this task have employed deep learning, such as convolutional neural networks (CNNs). CNNs are a specialized type of artificial neural network primarily tailored to processing and analyzing two-dimensional data, such as images. They excel at automatically extracting hierarchical patterns and features from visual data through layers of convolutional operations, enabling tasks like image recognition, object detection, and image segmentation with high accuracy (Figure 1) [18,19].

Their proficiency in extracting high-level hierarchical representations of the data through multi-step image processing has made them extremely successful [20]. Bhatnagar et al. [21] used three different CNN architectures in deciphering the Fashion-MNIST dataset for the classification of fashion article images. An accuracy of 92.54% was obtained by using a two-layer CNN with batch normalization and skip connections. CNNs have also been able to attain record beating outputs on extremely challenging datasets. Krizhevsky et al.’s [22] network contained five convolutional layers and three fully connected layers and was used on an ImageNet dataset. It achieved the best results ever reported: a test error rate of 15.3%. Their neural network included several novel and unusual features that increased performance, such as Relu nonlinearity and overlapping pooling.

AI has also been utilised in the sneaker world to predict the resale prices of sneakers, as rare pairs can be sold for much more than they were originally bought for. If a reseller knows how much a pair of sneakers can be sold for, they can determine whether it will be a worthwhile investment opportunity. Raditya and Hanafiah [23] used historical sneaker sales data from StockX to compare multiple algorithms, such as linear regression and random forest, to determine which one performed better in predicting sneaker resale prices. On evaluation, even though both algorithms produced similarly high R2 scores it was concluded that random forest performed better as its mean squared error was 30% lower when cross validation was applied. The use of AI and image classification is not just limited to the fashion industry. A closer look into the literature reveals a similar approach has been employed in healthcare where classifying medical images plays a vital role in aiding clinical care and treatment. For example, X-ray analysis is the best way to diagnose pneumonia [24] but classifying this disease from chest X-rays needs professional radiologists, which can be expensive in some regional areas [25]. This can lead to delays in diagnosis and treatment, which can cause negative patient outcomes. Thus, machine learning algorithms such as Support Vector Methods (SVMs), introduced by Cortes and Vapnik [26], and K-Nearest Neighbours (KNNs) have been used to aid with this task. Moreover, throughout the recent pandemic medical images were also used to effectively control COVID-19, from its spread to its mortality rate. Elaziz et al. [27] computed 961 image features from segmented regions of interest depicting chest X-ray images. After applying a feature selection, a KNN classification model was constructed and returned an accuracy of 96.1% when classifying between non-COVID-19 and COVID-19 cases. Nevertheless, they do have some drawbacks when used for classifying images according to Kermany et al. [28], such as time-consuming feature extraction and selection. Hence, as previously mentioned, CNNs are being employed more often in healthcare and they have even matched the performances of medical professionals in some studies. For example, CheXNet, a CNN with 121 layers trained on a dataset with over 100,000 frontal-view chest X-rays, attained a better performance than the average performance of four radiologists [29]. CNNs have the ability to automatically learn and compute discriminative features directly from the raw input data. Unlike manually engineered features, which can sometimes be hit-or-miss and heavily rely on human intuition, CNNs can effectively capture complex patterns and representations from the data themselves.

Another application of image classification is for security purposes. This is in part due to mobile phones becoming prevalent and being capable of detecting the biometric traits of humans. The recognition of an individual’s physical characteristics ensures an easy method of authentication, removing the inconvenience of traditional authentication methods likes identity cards and passports [30]. Fingerprints and facial recognition are some examples of biometric features. Image classifiers have been deployed in various ways. Yousif et al. [31] deployed their algorithm for classifying humans, animals, and empty frames from camera trap images with a user-friendly graphical user interface (GUI). This allowed ecologists to easily utilise the algorithms without detailed programming knowledge. Their GUI automatically divided the deployment images into sequences based on time stamps, color-coded the classified images, and was made available in Windows/Linux versions. Beede et al. [32] used a web application displaying their deep learning model’s predictions for diabetic retinopathy and diabetic macular edema. The images from the eye camera were sent to the algorithm in the cloud, and an assessment for the presence and severity of diabetic retinopathy, along with the existence or absence of diabetic macular edema, was returned in real time. This included a recommendation for whether patients should be screened and reviewed.

Based on this literature analysis, examples of adopting image classification techniques in the fashion world and other industries are numerous (Table 1). Yet it appears there are not many studies that address the use of these techniques, especially those using CNNs, both bespoke and pretrained, to identify whether sneakers being resold are genuine. There is a research gap in this field that this study is looking to address, and it will be of great benefit to ecommerce platforms in the sneaker reselling industry. Even though inputs and outputs are known, if asked how the outputs are selected algorithms cannot give reasoning that could be discussed within a human community. This is the case for many deep and machine learning models as they are virtually impossible to interpret. This is a significant issue in healthcare but less so in fashion as life is not at risk. In relation to sneakers, it may require a human to work alongside an algorithm on occasion to confirm why a particular item of clothing is counterfeit from closer inspection. This could be anything from a sneaker’s upper pattern direction to its logo size and positioning.

## 3. System Model: Methodology

The methodology applied in this work followed the traditional image processing pipeline (Figure 2). Two different algorithms were used on the dataset: CNN and SVM.

### 3.1. Data Selection

Real-life color jpeg formatted images of 3000 real and 2000 counterfeit sneakers were used for the classifications. They were provided by the Sole Supplier. The Sole Supplier made use of their image scraping system to build a massive volume of sneaker images. They also confirmed the fake sneaker images were counterfeit and the real sneaker images were genuine, as incorrect labelling would not be beneficial for the algorithm. One crucial point is that this work had a much lower number of images for model training: 500 for the SVM and 1000 for the CNNs. This was due to the online platform initially providing around 1000 photos of the same style of sneakers from different angles. The other multitude of images they had took too long to be provided, so they could not be used in this study. Therefore, the authors had to source more images of sneakers from an available Kaggle dataset. However, these were stock images which are unlikely to be used in a real-world setting for authentication. To try to overcome this, the authors took photos of their own authentic sneakers from various angles to add to the dataset even though this was a minimal number. The fake trainers had to be snipped/screenshotted from various websites, then augmented in different ways to create more. This did make it more difficult to know with 100% certainty that the images of the fake sneakers were of counterfeit goods.

All experiments in this study were conducted on a personal laptop with an Intel i3 processor. The dataset was uploaded to Google Colab as it provides free GPU/TPU resources from the browser for image classification. With Colab, it is possible to write and execute code, save and share analyses, and access powerful computing resources like pre-installed python libraries. The used python libraries in this work were matplotlib, scikit-learn, scikit-image, numpy, pandas, random, tensorflow, and keras.

### 3.2. Data Preprocessing

Data preprocessing has been found to increase classifier accuracy as it removes noise, detects outliers [36], reduces misclassification, and improves the recognition rate [37]. The dataset was checked to ensure it was balanced as too much or too little of one class can distort the classifiers’ performance. Data augmentation was utilised for all the CNNs. All images were gray-scaled as well as having histogram equalization applied and/or smoothing methods if required. They were then resized to 200 × 200 for the machine learning model and 300 × 300 for the deep learning model. Images were also resized accordingly to specified dimensions for use in various models. Pixel values were then normalized to obtain the dimensions in the same scale. These images and their subsequent labels were then separated into training and test data with a 70/30 split and a random state of 100.

### 3.3. Feature Extraction

Various feature extractions were trialled in conjunction with SVMs. SVMs utilise a set of flexible parameters to generate an optimal hyperplane between classes to attain accurate results [38]. They are less sensitive to a dataset’s dimensionality [39], making them beneficial for image classification. Feature extractors bridge the gap between the pixels and meaningful concepts such as color, shape, and texture so that the task of classifying patterns is made easy by a formal procedure. In this study, various methods of feature extraction were applied with SVMs, such as Sobel, Prewitt, Canny Edge Detection, and histogram of oriented gradient (HOG). Once feature extraction was performed on the dataset, it was then used to train, validate, and test the model. Features are aspects of an image that help to identify it. These feature extractors were trialled with the SVM image classifier to recognise the images of sneakers.

### 3.4. Deep Learning Model

The chosen model for this work was the CNN pioneered by LeCun et al. (1989), as deep learning models have made a great breakthrough in image classification tasks [33]. Feature extractors are not selected in a CNN as this is performed automatically. The number of convolutional layers, filters, kernel sizes, and strides were adjusted in the bespoke CNN to improve model performance. The initial weight and bias values of all feature maps were assigned with random normal distribution. Same-padding and max pooling were employed as well. As same-padding maximizes features captured in the image for optimal extraction, it has been found to significantly increase performance compared to valid-padding [40]. Max pooling can bring about faster convergence, superior invariant feature selection, and generalisation improvement [41]. Due the classification being binary, a “sigmoid” activation was used after vector flattening. As CNNs can be either built from scratch or fine-tuned with pretrained models, both strategies were utilised to see which performs best.

### 3.5. CNN Hyperparameter Tuning

Unlike model parameters, which are automatically optimised, hyperparameters are variables that need to be defined by the user before applying a learning algorithm to a dataset. They were utilised to try and improve the performances of CNNs. Since they are unknown, identifying the optimal hyperparameter values for a CNN is challenging and somewhat of a dark art requiring years of experience. Unfortunately, there is no fool-proof method for their configuration. The most important hyperparameters that need to be figured out are the number of the hidden unit, optimiser, learning rate, batch size, epochs, and kernel regulariser.

### 3.6. Evaluation

There are various measures used to evaluate the accuracy of classification models. Accuracy, loss, recall, precision, and F1-score were utilised to assess the classifiers’ performance. The accuracy metric is vital for a classifier’s global performance, even though it does not reveal how it performs for individual classes. F1-score, precision, and recall are much more sensitive metrics in this aspect. Confusion matrixes are regularly used to evaluate the performance of classification models. The predicted and observed classes are typically presented in a matrix table as shown in Figure 3. For a two-class problem, the confusion matrix produces four combinations of observed and predicted values. They are false positives (precision), false negatives, True Positives (Sensitivity) and True Negatives (Specificity).

## 4. Experimental Results

This section presents the experimental results of multiple deep and machine learning models on the dataset. It is separated into three parts: the SVM classifier, pretrained CNNs, and the bespoke CNN.

### 4.1. Results of SVM Classifier

First, an SVM classifier utilised six different preprocessing methods, two feature extraction methods, and three different kernels. The preprocessing methods applied were gray-scaling, histogram equalization, Gaussian smoothing, bilateral smoothing, and median smoothing. The feature extraction methods are Laplacian and HOG. On the other hand, the kernels are poly, linear, and RBF. Only the most accurate model’s confusion matrixes were displayed.

Table 2 shows that model 3 had the highest accuracy of 60%. This seemed to be due to changing the kernel. Model 3 had slightly lower values for precision, recall, and F1-score.

Figure 4 displays that there were only 90 correct classifications out of a possible 150. In total, 32 fake sneaker images were classified as real and 28 real sneaker images were classified as fake.

In the second experiment, we only applied a gray scale for preprocessing and different feature extractions with different kernels. Table 3 reveals that model 2 had the highest accuracy, which was similar to the top performing model from Table 2. Generally, adding the two additional feature extractors seems to have improved the accuracy. Precision, recall, and F1-score were slightly lower in this table than the previous ones as none reached above 0.49%.

Figure 5 shows that there were 88 correct classifications for this model and 62 that were incorrect.

### 4.2. Results of Pretrained CNN Result

We applied different CNN architectures like Alexnet, VGG16, and LeNet-5.

#### 4.2.1. Performance of AlexNet

Table 4 confirms that AlexNet model trials were not able to train properly despite numerous changes to the epochs, optimisers, and learning rates. The validation accuracies for Trials B–F did not increase and stayed at the same value throughout training, with Trial A yielding a poor result of 24.0%. Figure 6 illustrates what was occurring with the loss and accuracy during training and testing for one of these trials. Figure 6 displays one example of an AlexNet model trial in terms of loss and accuracy from Table 4. Trial D resulted in a poor validation accuracy of 50% like the others. Unfortunately, in this trial the model was not trained or fitted properly as the accuracy throughout testing stayed at 50%. The shape and trend of the loss for validation clearly shows that this model was not learning and was also suffering from overfitting. The potential reason for this may have been issues with either the model, the optimiser, or the hyperparameter value but as this was the case for all trials it would seem the AlexNet model was not applicable to this dataset.

#### 4.2.2. Performance of VGG-16

Table 5 reveals that most of the VGG-16 model trials produced high validation accuracy above 96%, except Trial A, which generated the poorest set of results with a validation accuracy of 33.4%. A combination of the Adam/Adamax optimiser, a learning rate of 0.00001, and kernel regularisers seemed to work well with VGG-16. Even though Trial B performed best, the graphs beneath Figure 7 display which specific VGG-16 model trials looked best in terms of accuracy and loss throughout training.

Figure 7 illustrates the best performing VGG-16 model trials in terms of accuracy during training and testing from Table 5. Trial B had a validation accuracy of 97.4%. Interestingly, unlike Trial E the accuracy for training started at 55.3% and that for testing started at 82.5%. As they did not start from the same point, it means the model had already learned before the training had started. However, after 1 epoch accuracy for training and testing were largely well matched. Convergence did happen on occasion for accuracy and loss. Loss for training and testing levelled out at the end, showing that this trial did not need to be run over further iterations. The loss graph also displayed that as the gap between the validation and training curves was small there was not much room for additional improvement with this model trial.

For Trial E, a validation accuracy of 96.3% was achieved and it was slightly overfit. Accuracy and validation accuracy were predominantly well matched throughout training and they both originated at the same point. There was a small drop in the validation accuracy at 5 epochs. There was significant improvement in accuracy for testing and training after 1 epoch. The curves for accuracy and loss were generally smooth. Loss for training and testing still seemed to be descending as the epochs increased, meaning the training was not finished and more was needed. The gap between the validation and training curves for loss showed there was not much area for improvement with this model trial. As the number of epochs increased, it appeared convergence would happen for accuracy. If this trial had been run over more iterations it may have brought some minimal improvement to the validation accuracy. In summary, Trial E with its kernel regularisation is the better VGG-16 model as accuracy for training and testing start at the same point even though its validation accuracy is slightly lower than that for Trial B.

#### 4.2.3. Performance of LeNet-5

Table 6 displays that LeNet-5 model Trials C, D, and E produced good validation accuracies above 78.0%. It seems the use of kernel regularisers with Nadam/Adagrad optimisers helped to boost validation accuracy in some trials. Adam and SGD optimisiers had a negative impact as they did not facilitate training and thus the validation accuracy of Trials A and B stayed at 50.0%. The graphs below (Figure 8) further explain which trials looked best throughout training in terms of loss and accuracy.

Figure 8 shows some of the top performing LeNet-5 model trials from Table 6 in terms of loss and accuracy. Trial C suffered from overfitting but resulted in a validation accuracy of 78.0%. This was lower than that of the best VGG-16 model trials from Table 5. The accuracy began around 58% for training and 50% for testing. Accuracy for training and testing were generally well matched even though there were some minor fluctuations in both and as the number of epochs increased the accuracies for training and testing were on course to converge. If the trial had been permitted to run over more iterations it may have given further improvement to validation accuracy. Loss for training and testing was beginning to flatten as the epochs increased, meaning the training was not finished and more was needed. The gap between the validation and training curves for loss showed this model trial could have been further improved.

Trial D achieved a validation accuracy of 81.4%. Interestingly, the accuracy for training started at 63% and the accuracy for testing started around 71%. As they did not start from the same area, this probably means the model had already started learning before the training had started. Although, after 1 epoch the accuracy for training and testing were mostly well matched. Convergence did occur occasionally for accuracy and loss at certain points, but this was earlier than expected. There were a few drops in accuracy for both testing and training. Loss for training was beginning to flatten more than testing as the epochs increased, meaning the model needed to be run over more iterations. If this trial had more epochs it may have brought some improvement to the validation accuracy. The loss graph did not exhibit overfitting but the gap between the validation and training curves displayed that there was room for this model trial to be bettered.

Trial E reached a validation accuracy of 80.5%. Accuracy and validation accuracy were predominantly well matched throughout training and they both started at 50%. At 11 epochs, there was a small drop in validation accuracy. Convergence for accuracy did happen at 14 epochs, so this trial may not have needed to be run over more iterations. However, loss for training and testing appeared to still be descending as the epochs increased, meaning the training was not finished and more was needed. The loss graph did not show overfitting but once again the space between the training and validation curves as the number of epochs increased reveal there was room for improvement with this model trial.

## 5. Results of Bespoke CNN

Table 7 reveals that for a bespoke CNN model Trials B, C, D, and E produced high validation accuracies above 87.7%. Trials B–D had five convolutional layers with varying numbers of neurons in the first dense layer. Trial E had half the number of filters of Trials B–D with 384 neurons in dense layer 1. The Adagrad optimiser produced high validation accuracy with the Nadam optimisers close behind. It seems that the more neurons are in the dense layer the higher the validation accuracy, and a lower number of filters gives a lower validation accuracy. The graphs in Figure 9 further illustrate which trials looked best throughout training in terms of loss and accuracy to give a clearer view of which trial performed best.

Although the validation accuracies for Trials B and C were high, they are not shown in Figure 9 as their accuracy and loss graphs were erratic throughout training. This was probably due to the Adam optimiser. Figure 9 shows that Trial D achieved a validation accuracy of 90.1%. Accuracy for training started at 60% and accuracy for testing started at approximately 62%, so around the same area. After 1 epoch, accuracy for training and testing were mainly well matched, especially at 10 epochs. However, there was a slight drop at 3 epochs for the test accuracy. Overlapping did happen on occasion with the accuracies and loss at certain points, but this was earlier than expected. Loss for training and testing started to flatten out at the end, showing the model was properly fitted. The loss graph does not present any overfitting and underfitting as curves are close together. If this trial had been run over more iterations it is unlikely it would have brought much improvement as the accuracies for training at testing were already well matched at 10 epochs.

Trial E reached a validation accuracy of 87.7%. Accuracy for training and testing started at approximately the same point around 50% and they were well matched throughout, except at 2 epochs, where the test accuracy was higher. Crossovers did happen on occasion for the accuracies and loss, but these were sooner than expected. Loss for training and testing was still descending as the number of epochs increased, meaning the training was not finished and more was needed. The loss graph does not appear to show much overfitting or underfitting as there was a minimal gap between the training and validation curves. However, if this trial was run over more iterations it is debatable whether it would have brought much improvement as the accuracies for training at testing were very close. Trial D seems to present the better model.

## 6. Discussion

The main aim of this work was to create a deep and machine learning model that would use images to identify whether a pair of sneakers for resale was genuine or not. The models would then be evaluated to see which performed better based on the following:RQ1. Does the machine or deep learning algorithm give better accuracy?

Research question 1 (RQ1) revolved around whether the machine or deep learning models chosen in this study gave better accuracy. It is evident from Table 2, Table 3, Table 4, Table 5, Table 6 and Table 7 that the deep learning models resulted in more accurate image classifiers. The best SVM model (Table 2) achieved 60.6%, whereas the best CNN model attained 97.4% in VGGI Trial B (Table 5). Even the worst performing CNN model with an accuracy of 81.4% in LeNet-5 Trial D (Table 6) comprehensively bested all the SVM models. Figure 5 and Figure 6 also display that the maximum number of correct classifications was 91 with false positives and negatives being higher than expected.Q2. Which convolutional neural network architecture gives the highest accuracy?

Research question 2 explored which convolutional neural network gave the highest accuracy. The results from Table 4, Table 5, Table 6 and Table 7 confirm that the deep architecture of VGG-16 achieved the highest accuracy of 97.4%. The shallower layout of the bespoke five-layered CNN was a close second with an accuracy of 96.3% and LeNet-5 attained an accuracy of 81.4%. Unexpectedly, AlexNet was unable to be suitably trained.

### 6.1. Performance of the CNN versus SVM

On analysis of this work’s results, it is clear to see that CNN models gave a much a higher accuracy than those that used SVM. Table 4, Table 5, Table 6 and Table 7 show that all the CNN models except AlexNet were able to achieve accuracies that far surpassed those of the SVM classifiers. Even the worst performing LeNet-5 model with an accuracy of 78.0% still comfortably outperformed the best SVM model by nearly 18%. This is unsurprising, as CNNs have been found to have the ability to recognise objects better than machine learning models. This is mainly thanks to increased computer power, which means CNNs have more data available to train them.

The best performing CNN and SVM are presented in Table 8. In this table, VGG-16 is a large network which takes advantage of very few convolutional filters in all its layers. It reached an impressive accuracy of 97.4% (Table 5), demonstrating that a bigger CNN size is beneficial for improving image classification accuracy. One drawback of VGG-16 is that it took 7–8 h to train on only 1000 images even when utilising a GPU with a small number of epochs, as Figure 7 illustrates. It is therefore essential to have access to powerful GPUs or TPUs for prolonged periods when training CNNs.

Moreover, the LeNet-5 model trials (Table 6) may have been able to achieve better performance if they were able to run over more iterations as the training and testing accuracies were close to convergence and loss curves had yet to flatten (Figure 8). Regrettably, extended GPU usage was not available for this project as the author did not have access to Colab Pro or Colab Pro+, so smaller epochs had to be utilised to reduce runtime. This may have had an adverse effect on the validation accuracy. Therefore, even though deep learning can achieve considerably better accuracy it does need increased processing power, which can be costly and must be taken into consideration.

The only CNN model that yielded poor results was AlexNet (Table 6). Guo et al. [42] suggested AlexNet’s two primary shortcomings are that it requires a fixed image resolution and there is no obvious insight into why it performs so well. Despite multiple optimisers being tried, such as Adam, Adagrad, Adamax, Adadelta, Nadam, and RMSprop, learning rate changes, data augmentation during training, and batch size adjustments, on every occasion the testing accuracy did not improve as shown by Figure 6. This means the model was not training/learning properly for some reason, which is surprising as even though AlexNet is shallower than VGG-16 it has still been found to perform as well as it. One potential reason the AlexNet model may not have learned could be that it was affected by noise, but as the VGG-16, LeNet-5, and Bespoke CNN models were able to be trained on the dataset that hypothesis appears unlikely. Another explanation might be AlexNet’s use of batch normalization between each convolutional and pooling layer, as on closer examination it was clear to see the other CNN architectures in this project did not utilise this. Batch normalization aims to improve CNN training speed and stability, yet there is not full unanimity on why this technique is effective.

The SVM results demonstrated that the best performing SVM model had an accuracy of 60% with only the use of gray-scaling for image preprocessing and HOG for feature extraction. Unexpectedly, despite the addition of numerous smoothing techniques and more edge detection methods, an accuracy above 60% could not be reached. Some of the top performing models in Table 2 and Table 3 did manage to come close to this number. This disappointing accuracy may have again been caused by the size of the dataset as explained above. There also seemed to be no real discernible pattern for which kernel would generate better model performance. With some models linear performed best, while for others RBF and poly were best.

### 6.2. Comparisons with Related Works

As the dataset used in this study was balanced, accuracy was a good measure to evaluate the performance of the deep and machine learning models. It is evident from all the SVM results that SVM classifiers achieved a low accuracy when classifying the sneaker images as 60.6% was the highest achieved. In comparison to Greeshma and Sreekumars [16] and Greeshma and Sreekumars [33], SVM classifiers had an accuracy 87.4%. However, they did use the Fashion-MNIST dataset, which contains 10,000 images.

In terms of other evaluation methods, precision and recall for the SVM classifiers are of particular significance when it comes to classifying the sneakers. This is because they help to minimise the chance of false negatives—fake sneakers being classified as real—false positives—real sneakers being classified as fake—and more (Figure 5 and Figure 6). Precision and recall values should ideally be as close to one as possible but unfortunately, as Table 2 and Table 3 clearly show, no SVM models were able to come close to this. The usage of gray-scaling, median smoothing with HOG and gray-scaling, Gaussian smoothing, and histogram equalization with HOG was able to achieve mediocre precision and recall values. Interestingly, these were both with the poly kernel. In a practical sense, this would mean a lot of the sneakers identified as false positives and negatives would need to be checked by hand if this classifier was employed in the industry. This would defeat the purpose of using the AI system as it would not speed up the authentication process. Kart [43] used SVM in conjunction with HOG to classify images into two categories: men’s and women’s attire. They had an unbalanced training dataset of 753 images and 453 for testing. Their model achieved a precision of 0.79 and a recall of 0.78 with the global and nine grids feature extraction approach. These superior results may have been due to the larger dataset used or the fact that in this study real and stock images were utilised in training. Furthermore, Kart [43] only utilised the RBF kernel for their SVMs because it could be used as both linear and sigmoid kernels by adopting specific parameters, whereas this work made use of poly, linear, and RBF kernels.

In comparison with other studies, Bhatnagar et al. [21] achieved an accuracy of 92.5% with a bespoke two-layered CNN on the Fashion-MNIST dataset. The resultant 90.1% accuracy of the five-layered bespoke CNN in this project is quite similar. However, Bhatnagar et al. [21] used the much larger Fashion-MNIST dataset and their CNN was trained over considerably more epochs. Obaid and Jasim [44] also used VGG-16, AlexNet, and a bespoke five-layered CNN to classify the 70,000 images contained within the Fashion-MNIST dataset. A total of 60,000 images were for training and 10,000 for testing. They augmented their dataset images in three ways: rotation, acute angulation, and tilting. This created a larger dataset comprising 180,000 training images and 30,000 testing images. The 5 layered CNN attained a 94% accuracy, VGG-16 93% and 92% for the AlexNet. The bespoke five-layered CNN in this study performed marginally worse, with an accuracy of 90.1% (Table 7), but the VGG-16, with an accuracy of 97.4% (Table 5), performed slightly better. As explained earlier, as the AlexNet model in this report did not train properly a legitimate comparison could not be made. It does appear that data augmentation enhances the accuracy of CNNs as there are more training data for the classifier to learn from.

## 7. Conclusions and Future Work

### 7.1. Conclusions

The sneaker reselling industry is worth billions of GBP, but unfortunately it has a real issue with counterfeit goods which are being made to look more and more genuine. These can ruin buyer experiences and damage an online platform’s reputation. The current human sneaker authentication process is time consuming, at risk of error, and brings about environmental challenges. The aim of implementing an artificial intelligence system into the authentication process was to bring about improvements for both consumers and businesses when purchasing/selling sneakers. The results of this work clearly show that a highly accurate image classifier (97.4%) can be created from a VGGI CNN and that it should be able to distinguish between genuine and counterfeit sneakers much quicker and more efficiently. This would be of great benefit to online buyers because it would give them peace of mind that the sneakers they may have spent a large amount of money on are 100% authentic. Ecommerce platforms would also be at an advantage as they could divert human resources to other parts of their business if needed. In order to keep the model up to date and relevant with the latest sneaker styles and colors, images of real and counterfeit sneakers need to be continuously added to the model once they are released so it can be trained consistently. It may also be beneficial to set up separate CNNs for each sneaker style. Lastly, to potentially obtain the most benefit from the AI system it should be used in conjunction with human sneaker authentication until its accuracy can be further increased.

### 7.2. Future Work

One opportunity for future investigation would be the utilisation of a larger dataset that would contain more images of various sneaker brands, styles, and colors. The dataset used for this study was relatively small compared to similar studies. Having a larger dataset with numerous sneaker images would help the model classify the authenticity of the sneakers better and mean it is less likely for it not to recognise a wide variety of sneakers. This would vastly improve the model’s performance and its industry usefulness. However, larger datasets require more powerful systems for data processing so this needs to be considered.

Another aspect that can be looked at by others going forward is the use of other machine learning techniques for image classification on this specific dataset. There are various ones to choose from, such as K-Nearest Neighbour, an algorithm which generates groups of objects by classifying their features based on their nearest neighbour; decision trees; and random forest. Furthermore, the use of Shi–Tomasi corner detection with both scale-invariant feature transform and sped-up robust features as the method of image feature extraction can also be looked at. This could be then compared to edge detection, which was used in this project to assess which method gives better accuracy in an image classifier.

A further area for future research would be the use of KerasTuner for hyperparameter tuning. This process finds the optimal learning rate and the number of hidden units in a convolutional layer to give the CNN model its lowest loss and highest accuracy. The benefit of this is that the optimal number of hidden units and learning rate do not need to be guessed as the best performing combination is automatically chosen. Moreover, the author would suggest trialling more pretrained CNNs on the dataset going forward, such as Google LeNet, ResNet50, Inceptionv3, and EfficientNet.

## Figures and Tables

**Figure 1 sensors-24-03030-f001:**
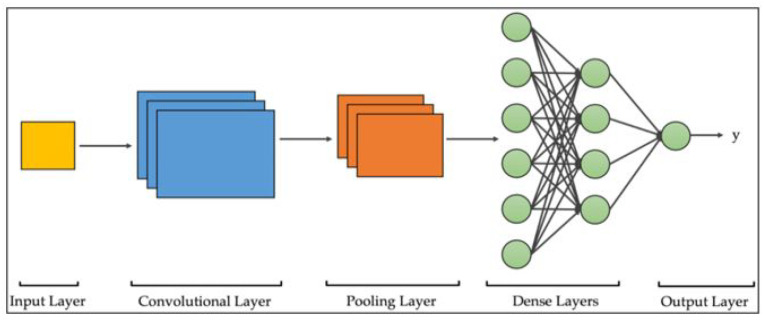
Pipeline of general CNN architecture.

**Figure 2 sensors-24-03030-f002:**
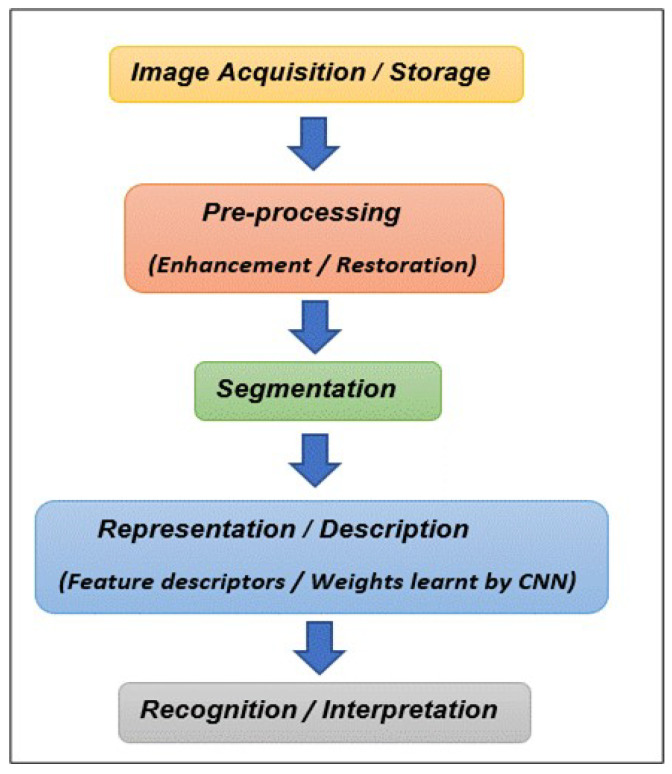
Image processing pipeline.

**Figure 3 sensors-24-03030-f003:**
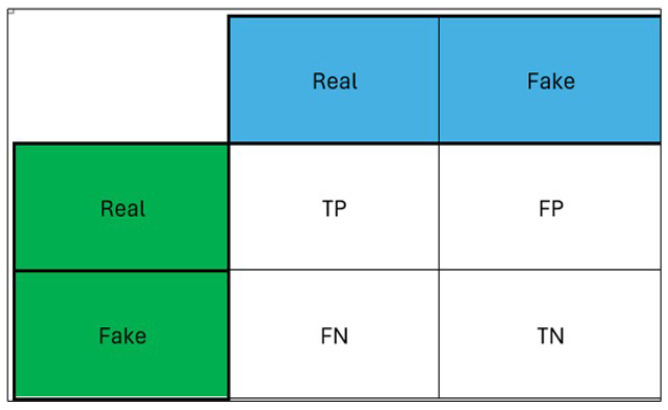
Confusion matrix.

**Figure 4 sensors-24-03030-f004:**
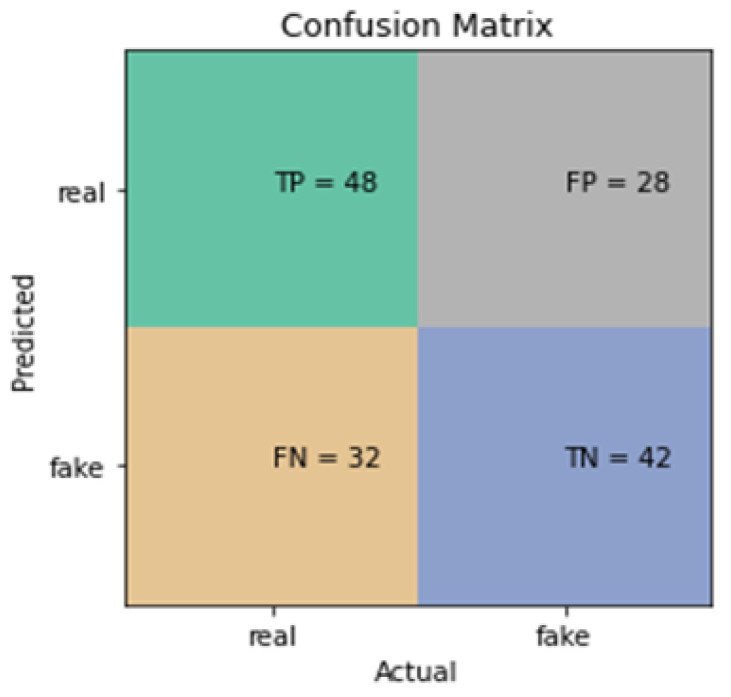
Confusion matrix for most accurate SVM model from Table 2.

**Figure 5 sensors-24-03030-f005:**
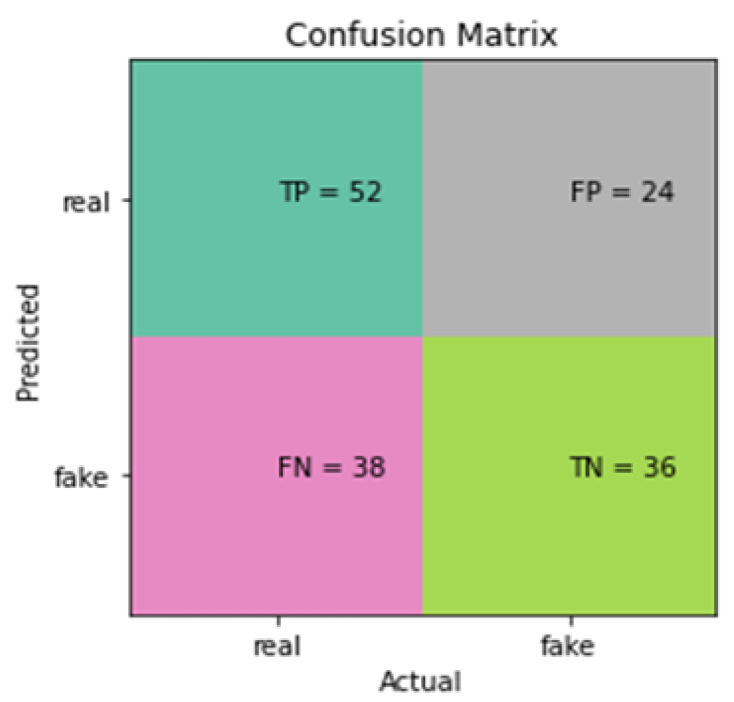
Confusion matrix for most accurate model from Table 4.

**Figure 6 sensors-24-03030-f006:**
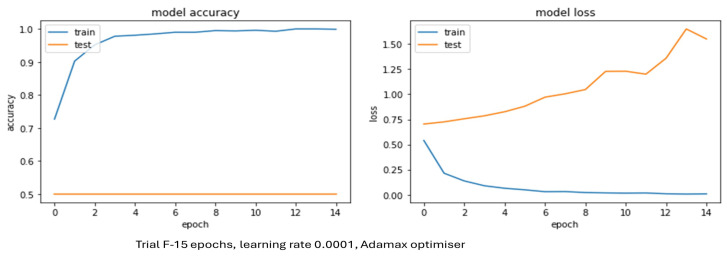
AlexNet model accuracy and loss for training and validation.

**Figure 7 sensors-24-03030-f007:**
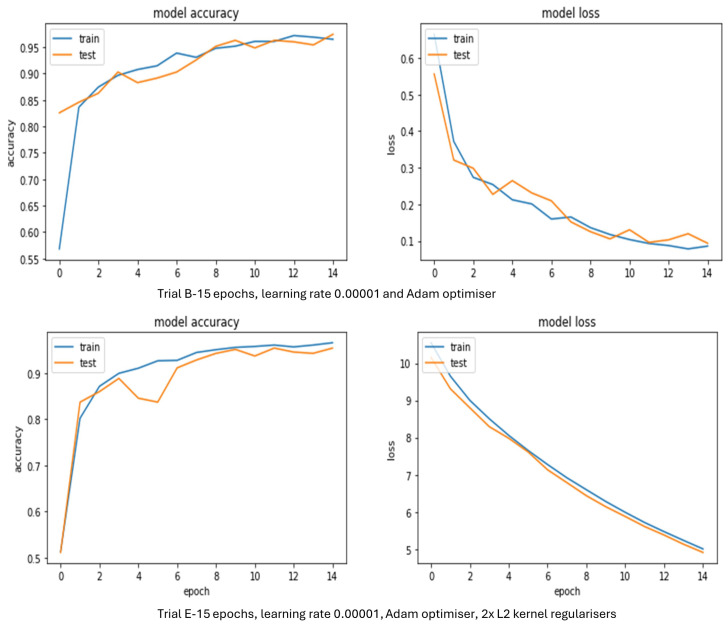
VGG-16 model accuracy and loss evolution during training and validation.

**Figure 8 sensors-24-03030-f008:**
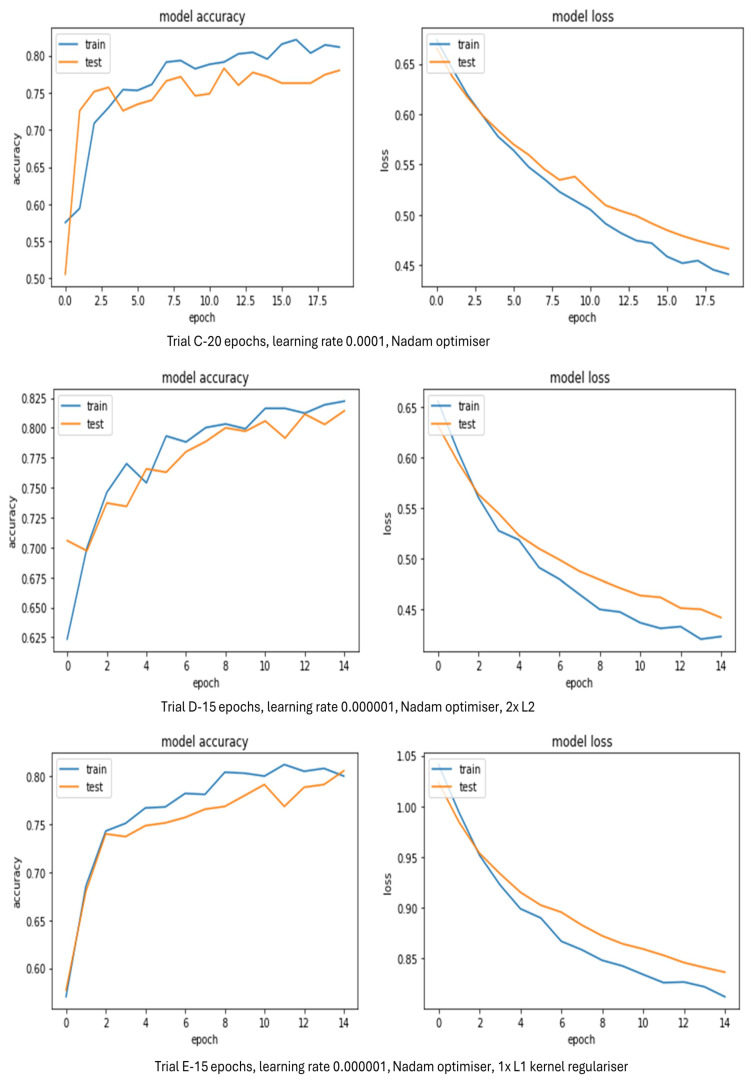
LeNet-5 model accuracy and loss evolution during training and validation.

**Figure 9 sensors-24-03030-f009:**
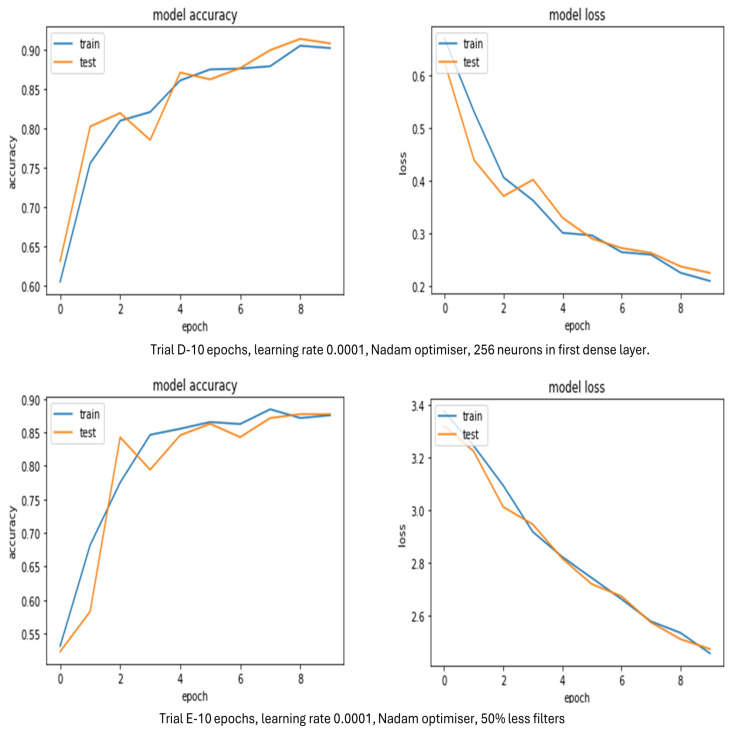
Bespoke CNN model accuracy and loss evolution during training and validation.

**Table 1 sensors-24-03030-t001:** Summary of literature review.

Citation	Type	Approach	Apps	Accuracy
[33]	Image Classifier	SVM	Clothes	86.8%
[16]	Image Classifier	SVM	Clothes	87.4%
[17]	Image Classifier	SVM	Clothes	85.6%
[27]	Image Classifier	KNN	Medical X-ray chest	96.1%
[21]	Image Classifier	CNN	Clothes	92.54%
[29]	Image Classifier	CNN	Medical X-ray chest	-
[34]	Image Classifier	CNN	Breast cancer	99.6%
[35]	Image Classifier	CNN	COVID-19 X-ray	94.5%
[23]	Predictor	LR and RF	Sneaker resale price prediction	94.5%

**Table 2 sensors-24-03030-t002:** Multiple smoothing methods with HOG and Laplacian.

Model	Preprocessing	Feature Extraction	Kernel	Accuracy (%)	Precision	Recall	F1-Score
1	gray-scaling, histogram equalization, Gaussian smoothing, bilateral smoothing, median smoothing	Laplacian, HOG	poly	49.3	0.52	0.52	0.52
2	gray-scaling, histogram equalization, Gaussian smoothing, bilateral smoothing, median smoothing	Laplacian, HOG	linear	54.0	0.52	0.52	0.52
3	gray-scaling, histogram equalization, Gaussian smoothing, bilateral smoothing, median smoothing	Laplacian, HOG	RBF	60.0	0.51	0.51	0.51

**Table 3 sensors-24-03030-t003:** Gray-scaling with HOG, sobel_h, and prewitt_v.

Model	Preprocessing	Feature Extraction	Kernel	Accuracy	Precision	Recall	F1-Score
1	gray-scaling	sobel_h, prewitt_v, HOG	linear	58.6%	0.49	0.49	0.49
2	gray-scaling	sobel_h, prewitt_v, HOG	poly	59.3%	0.48	0.48	0.48
3	gray-scaling	sobel_h, prewitt_v, HOG	RBF	55.3%	0.47	0.47	0.47

**Table 4 sensors-24-03030-t004:** AlexNet model performance.

Trial	Epochs	Learning Rate	Opt	Train Accuracy	Train Loss	Validation Accuracy
A	10	0.1	Adam	69.9%	11.9	24.0%
B	10	0.000001	Adam	89.2%	0.2	50.0%
C	15	0.001	RMSprop	95.3%	0.2	49.4%
D	15	0.001	Adagrad	62.5%	2444.3	50.0%
E	15	0.0001	Nadam	98.4%	1349.1	50.0%
F	15	0.0001	Adamax	99.9%	0.1	50.0%

**Table 5 sensors-24-03030-t005:** VGG-16 model performance.

Trial	Epochs	Learning Rate	Opt	Train Accuracy	Train Loss	Validation Accuracy	Validation Loss
A	10	0.1	Adam	31.2%	nan	33.4%	nan
B	15	0.00001	Adam	96.1%	0.08	97.4%	0.09
C	15	0.00001	Adagrad	97.2%	0.13	96.6%	0.15
D	15	0.00001	Adam	95.8%	2.49	96.3%	2.44
E	15	0.00001	Adam	95.8%	2.49	96.3%	2.44

**Table 6 sensors-24-03030-t006:** LeNet-5 model performance.

Trial	Epochs	Learning Rate	Opt	Train Accuracy	Train Loss	Validation Accuracy	Validation Loss
A	10	0.01	Adam	49.2%	0.71	50.0%	0.70
B	10	0.000001	Adam	56.6%	0.72	50.0%	0.73
C	20	0.0001	Adagrad	81.1%	0.44	78.0%	0.46
D	15	0.000001	Adam	82.3%	0.42	81.4%	0.44
E	15	0.000001	Adam	80.0%	0.81	80.5%	0.84

**Table 7 sensors-24-03030-t007:** Bespoke CNN model performance.

Trial	Epochs	Learning Rate	Opt	Train Accuracy	Train Loss	Validation Accuracy	Validation Loss
A	15	0.01	RMS Prop	48.3%	0.72	50.0%	0.71
B	30	0.01	Adagrad	99.8%	0.01	95.0%	0.17
C	25	0.01	Adagrad	99.9%	0.01	96.3%	0.11
D	10	0.0001	Nadam	90.2%	0.21	90.1%	0.22
E	10	0.0001	Nadam	87.5%	2.45	87.7%	2.47

**Table 8 sensors-24-03030-t008:** Comparison of SVM with CNN architectures.

	SVM	CNN			
		**AlexNet**	**VGG-16**	**LeNet-5**	**Bespoke**
Accuracy (%)	60	50	97.4	81.4	96.5

## Data Availability

Available upon request.

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
