# Peer review of "Image Classifier for an Online Footwear Marketplace to Distinguish between Counterfeit and Real Sneakers for Resale"

_sensors, 2024, doi:10.3390/s24103030_

Round 1

Reviewer 1 Report

Comments and Suggestions for Authors

In this paper, the author design and develop an image classifier to help the sole suppliers sneaker authenticators to distinguish between real and counterfeit sneakers so less human inspection is required. To speed up the authentication process of shoes, the author uses support vector machines and convolutional neural networks to classify images of fake and real sneakers and then their accuracies were compared to see which performed better. The results showed CNNs performed much better at this task than the SVMs with some accuracies over 95%. Therefore, a CNN is well equipped to be a sneaker authenticator and will be of great benefit to the reselling industry. However, there are still some issues in this article that need to be explained clearly. Here below are my comments.

[1]      Figures 1, 2, 6, and 7 are too blurry and need to be redrawn. Additionally, there is a spelling error on line 63.

[2]      There is too much content in the introduction to introduce "bots", which is not closely related to the main content of the article. The content about "bots" should be deleted, with a focus on using artificial intelligence for shoe recognition.

[3]      The main problem with this study is the lack of innovation, as it did not adjust the model architecture specifically for the task of shoe identification and simply used other people's models.

[4]      Comparing SVM and CNN is an outdated topic, and designing models specifically for identifying the authenticity of shoes is more meaningful.

[5]      The tables in the paper should be in the form of tables, rather than using representations similar to images.

[6]      The number of datasets tested in the paper is a bit too small and needs to be tested on some larger or standard datasets.

Comments on the Quality of English Language

Language can be further improved.

Author Response

All the comments have been addressed one by one in the attached file.

Reviewer 2 Report

Comments and Suggestions for Authors

This work propose a image classifier for sneaker data. The method is reasonable and results are well presented. However, I still have some concerns:

1) The quality (resolution) of figures are quite low, especially for Figure 1 and 2. Please ensure a high-resolution figure to clearly deliver your information.

2) My major concern is regarding the novelty. Does the author aim to propose a novel method or aim to provide benmarks for sneaker data? Please clarify your major conclusion clearly.

3) The presentation of results is also not clear. Please report the test results in the Tables instead of only describing in the manuscript.

4) Please proofread the manuscript to fix typos.

Comments on the Quality of English Language

See the comments above.

Author Response

(The authors gave the same response as above.)

Reviewer 3 Report

Comments and Suggestions for Authors

The aim of the research that is reported is to develop and evaluate a classifier to differentiate between genuine and fake limited edition sneakers. A CNN and an SVM were trained for the task. The best CNN had a success rate of 97%.

The paper has a section to introduce the problem. This is followed by a brief literature review. The methodology is described, not always clearly and not always exhaustively. Results from a number of experiments are presented. The experiments are not always convincing. The results are discussed. There is a conclusions section that summarises the paper and makes suggestions for further work.

There are some questions that need to be addressed:

Why use the SVM classifier, are there any newer, more powerful classifiers?

Why do you restrict the feature detectors to HOG plus edge detectors? There are better feature detectors.

Why restrict the experiment to three specific CNN architectures plus the bespoke?

As per my comments below, many sections of the paper need further development. Very minor editing to the English is also required. 

Section 1

The use-case is not explained very clearly, the jump between online sales by manufacturers (which wouldn’t need verification) and online reselling by individuals (which does need verification) is not made.

Section 2

The literature review seems to concentrate on applications that have used a single image database. Has this excluded other applications? Some of the comments I’ve made suggest that the authors do not appreciate the properties of CNNs.

Line 85

Need to rewrite the definition of CNN, their uses are more widespread than this would suggest.

Line 125

Isn’t the advantage of using a CNN as a feature extractor is that it can be trained to compute the most discriminatory features, whereas human engineered features are very hit or miss? Complicated, expensive and slow are not reasons for this.

Line 128

Is the mention of mobile phones here strictly accurate? Most CNN models are very large, so unsuited to mobile devices. Reduced models can be derived, but they usually have a reduced performance.

Line 145

“not many studies that address the use of these techniques” Is this true?!! Putting “application of CNN” into Google gives 113,000,000 results.

Line 151

That networks cannot explain themselves is an incorrect statement. It is entirely possible to have the CNN highlight the regions of the image that are important in making decisions.

Line 160

“Official” image processing pipeline? Traditional may be a better description.

Lines 182, 183, 184

How were pixel values normalised?

What does “get the dimensions in the same scale” mean?

What is “a random state of 100”?

Line 193

The lit review stated that one of the better classifiers used HOG and LBP (ref 17). So why didn’t this study use that?

Line 231

What is the “threshold” of a confusion matrix? Are you referring to a threshold in the classifier?

Table 2

Would benefit from horizontal lines to separate the three models.

Do I interpret the table correctly – the best results are obtained when using all six pre-processing methods, including three variants of smoothing?

This and table 3 do illustrate the hit or miss nature of engineered feature extraction: it is impossible to know that these are the best possible results that can be achieved using manual features and an SVM classifier without trying every feature detector.

Line 256

Is there any justification for choosing these three models?

Section 6

The results you present support the argument that a CNN gives better results than the SVM+engineered features. But would other features have given better results than you have presented. I’m worried that you didn’t implement the method of reference 17 that your lit review says gave the best results in that application.

The comments regarding the dataset should have appeared in the methodology section.

Line 437

The paper has said throughout that AlexNet failed to learn. But here you say that it did learn, once the batch normalisation was removed. I suggest you do this experiment and report these results instead of the negative result that’s in the paper.

Line 452

Ref 17, as well as using different data and more of it, used different feature extractors.

Line 466

You write about balancing as though the amount off training and testing data should be the same. Is this what you mean? Normally, balanced refers to having the same numbers of genuine and fake sneakers.

Comments on the Quality of English Language

Very minor corrections, such as missing apostrophes, are needed.

Author Response

(The authors gave the same response as above.)

Round 2

Reviewer 1 Report

Comments and Suggestions for Authors

The author has made significant revisions to the previous comments, resulting in a significant improvement in the quality of the paper. However, there are still some issues that need to be addressed in the paper. Here below are my comments.

(1)The font in Figures 1, 4, and 5 is not very clear.

(2)I recommend that the authors consider open-sourcing the dataset.

(3)In the experimental results section, to demonstrate that CNN performs much better than SVM in this task, there needs to be a table that compares the performance metrics of SVM classifier, pretrained CNNs and bespoke CNN together.

Author Response

Hi there,

All the comments have been addressed in the attached file

Reviewer 2 Report

Comments and Suggestions for Authors

The authors have addressed most of my technical comments. Please polish your presentation when preparing the final version.

Author Response

Hi there,

All the comments have been addressed in the attached file.

Reviewer 3 Report

Comments and Suggestions for Authors

The authors have improved the presentation of the paper. They have given responses to the queries I made in my original review that are more or less satisfactory.

Comments on the Quality of English Language

Some minor errors remain, at the level of incorrect or missing apostrophes.

Author Response

(The authors gave the same response as above.)
